# Power-law efficient neural codes provide general link between perceptual bias and discriminability

**Michael J. Morais & Jonathan W. Pillow**
Princeton Neuroscience Institute & Department of Psychology
Princeton University
`mjmorais, pillow@princeton.edu`

## Abstract

Recent work in theoretical neuroscience has shown that efficient neural codes, which allocate neural resources to maximize the mutual information between stimuli and neural responses, give rise to a lawful relationship between perceptual bias and discriminability in psychophysical measurements (Wei & Stocker 2017, [1]). Here we generalize these results to show that the same law arises under a much larger family of optimal neural codes, which we call *power-law efficient codes*. These codes provide a unifying framework for understanding the relationship between perceptual bias and discriminability, and how it depends on the allocation of neural resources. Specifically, we show that the same lawful relationship between bias and discriminability arises whenever Fisher information is allocated proportional to any power of the prior distribution. This family includes neural codes that are optimal for minimizing $L_p$ error for any $p$, indicating that the lawful relationship observed in human psychophysical data does not require information-theoretically optimal neural codes. Furthermore, we derive the exact constant of proportionality governing the relationship between bias and discriminability for different choices of power law exponent $q$, which includes information-theoretic ($q = 2$) as well as "discrimax" ($q = 1/2$) neural codes, and different choices of decoder. As a bonus, our framework provides new insights into "anti-Bayesian" perceptual biases, in which percepts are biased away from the center of mass of the prior. We derive an explicit formula that clarifies precisely which combinations of neural encoder and decoder can give rise to such biases.

## 1  Introduction

There are relatively few general laws governing perceptual inference, the two most prominent being the Weber-Fechner law [2] and Stevens' law [3]. Recently, Wei and Stocker [1] proposed a new perceptual law governing the relationship between perceptual bias and discriminability, and showed that it holds across a wide variety of psychophysical tasks in human observers.

Perceptual bias, $b(x) = \mathbb{E}[\hat{x}|x] - x$, is the difference between the average stimulus estimate $\hat{x}$ and its true value $x$. Perceptual discriminability $D(x)$ characterizes the sensitivity with which stimuli close to $x$ can be discriminated, equivalently the just-noticeable difference (JND); this is formalized as the stimulus increment $D(x)$ such that the stimuli $x + \eta D(x)$ and $x - (1 - \eta)D(x)$ (for $\eta$ between 0 and 1) can be correctly distinguished with probability $\geq \delta$, for some value of $\delta$. Note that by this definition, lower discriminability $D(x)$ implies higher sensitivity to small changes in $x$, that is, improved ability to discriminate.

The law proposed by Wei and Stocker asserts that bias and discriminability are related according to:

$$b(x) \propto \frac{d}{dx} D(x)^2 \qquad (1)$$

where the right-hand-side is the derivative with respect to $x$ of the discriminability squared. The relationship is backed by remarkably diverse experimental support, crossing sensory modalities,

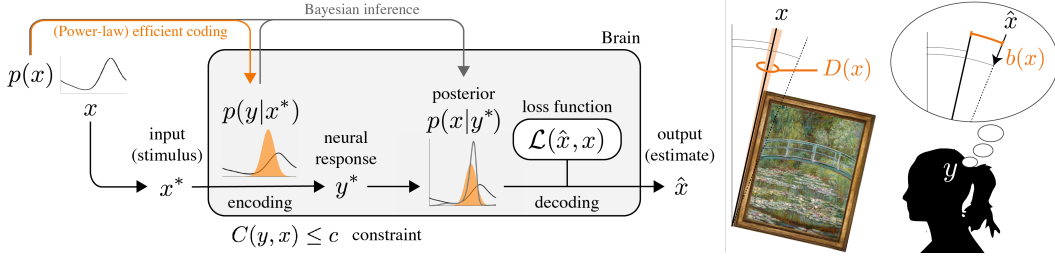

Figure 1: *(Left)* Schematic of Bayesian observer model under power-law efficient coding. On each trial, a stimulus $x^*$ is sampled from the prior distribution $p(x)$, and encoded into a neural response $y^*$ according to the encoding distribution $p(y|x^*)$. Inference involves computing the posterior $p(x|y^*) \propto p(y^*|x)p(x)$, and the optimal point estimate $\hat{x}$ minimizes the expected loss $\mathbb{E}_{p(x|y^*)}[\mathcal{L}(\hat{x}, x)]$. Power-law efficient coding stipulates that the encoding distribution $p(y|x)$ has Fisher information proportional to $p(x)^q$ for some power $q$. Thus the prior influences both encoding (via the Fisher information) and decoding (via its influence on the posterior). *(Right)* Intuitive example of bias and discriminability: adjusting a crooked picture frame. The stimulus $x$ represents the angle off of vertical. Discriminability $D(x)$ measures the minimum adjustment needed for the observer to detect that it became better (or worse). Bias $b(x)$ measures the offset of the estimated angle $\hat{x}$ from the true angle, in this case the overestimation of the crookedness. Adapted with edits from [1].

stimulus statistics, and even task designs. At the heart of this experiment-unifying result is the Bayesian observer model, flexibly instantiating perception as Bayesian inference in an encoding and decoding cascade with a structure optimized to statistics in the natural environment [4, 5].

Wei and Stocker derived their law under the assumption of an information-theoretically optimal neural code, which previous work has shown to hold when Fisher information $J(x)$ is proportional to $p(x)^2$, the square of the prior distribution [6–8]. A critical follow-up question is whether this condition is necessary for the emergence of the perceptual law. Does the perceptual law *require* information-theoretically optimal neural coding, or does the same bias-disriminability relationship arise from other families of (non-information-theoretic) optimal codes? Here we provide a definitive answer to this question. We use a Bayesian observer model to generalize the Wei-Stocker law beyond information-theoretically optimal neural codes to a family that we call power-law efficient codes. These codes are characterized by a power-law relationship between Fisher information and prior, $J(x) \propto p(x)^q$, for any exponent $q > 0$. Critically, we show that this family replicates all key results in the original Wei and Stocker model.

We first review the derivation of the Wei & Stocker result governing the relationship between bias and discriminability (Section 2). We then develop a generic variational objective for power-law efficient coding that reveals a many-to-one mapping from objective to resultant optimal code (Section 3). We use this objective to derive a nonlinear relationship between bias and discriminability that, in the limit of high signal-to-noise ratio (SNR), reproduces the Wei & Stocker result for all power-law efficient codes, with an analytic expression for the constant of proporationality (Section 4). In simulations, we explore a range of SNRs and power-law efficient codes to verify these results, and examine a variety of decoders including posterior mode, median, and mean estimators (Section 5), demonstrating the universality of the bias-discriminability relationship across a broad space of models.

## 2   The Wei & Stocker Law

The perceptual law proposed by Wei and Stocker can be seen to arise if perceptual judgments arise from a Bayesian ideal observer model with an appropriate allocation of neural resources. Perceptual inference in the Bayesian observer model (Fig. 1) consists of two stages: (1) *encoding*, in which an external stimulus $x$ is mapped to a noisy internal representation $y$ according to some encoding distribution $p(y|x)$; and (2) *decoding*, in which the internal representation $y$ is converted to a point estimate $\hat{x}$ using the information available in the posterior distribution,

$$p(x|y) \propto p(y|x)p(x), \tag{2}$$

which (according to Bayes' rule) is proportional to the product of $p(y|x)$, known as the likelihoood when considered as a function of $x$, and a prior distribution $p(x)$, which reflects the environmental

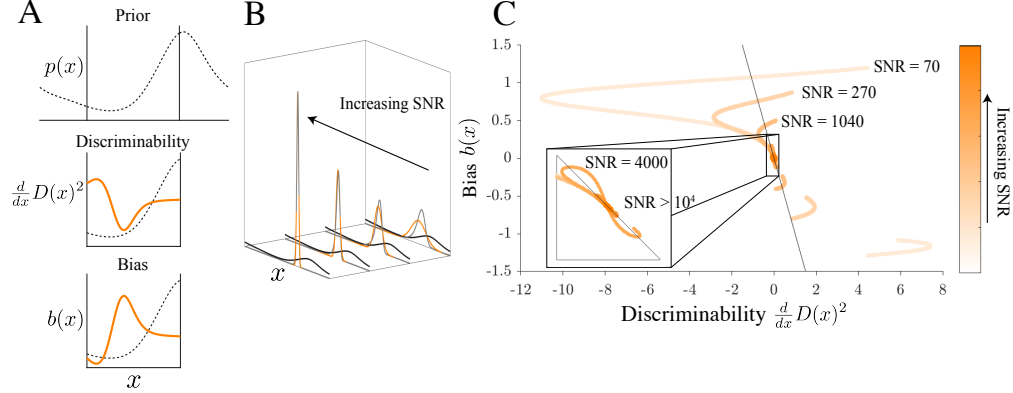

Figure 2: The high-SNR regime within which the bias-discriminability relationship linearizes, under the same sample prior as in Figure 1. **(A)** Schematic illustration of how prior (top) relates to discriminability (middle) and bias (bottom). **(B-C)** Increasing SNR $k$ narrows the likelihood function (orange) and posterior (gray) relative to the prior (black), and makes the posterior more Gaussian. **(D)** The bias-discriminability relationship has arbitrary curvature at low-SNR, but converges to a line with known slope in the high-SNR limit.

stimulus statistics. Technically, the Bayes estimate is one that minimizes an expected loss under the posterior: $\hat{x}_{Bayes} = \arg\min_{\hat{x}} \int dx\, p(x|y) L(x, \hat{x})$, for some choice of loss function (e.g., $L(x, \hat{x}) = (x - \hat{x})^2$, which produces the "Bayes least squares estimator").

Optimizing the encoding stage of such a model involves specifying the encoding distribution $p(y|x)$. Intuitively, a good encoder is one that allocates neural resources such that stimuli that are common under the prior $p(x)$ are encoded more faithfully than stimuli that are uncommon under the prior. Recent work from several groups [6–9] has shown that his allocation problem can be addressed tractably in the high-SNR regime using Fisher Information, which quantifies the local curvature of the log-likelihood at $x$:

$$J(x) = \mathbb{E}_{y|x}\left[ - \frac{\partial^2}{\partial x^2} \log p(y \mid x) \right]. \tag{3}$$

In the high-SNR regime, Fisher information provides a well-known approximation to the mutual information between stimulus and response: $I(x, y) \approx \frac{1}{2} \int dx\, p(x) \log J(x) + const.$ This relationship arises from the fact that asymptotically, the maximum likelihood estimate $\hat{x}$ behaves like a Gaussian random variable with variance $\sigma^2 = 1/J(x)$ [9, 7, 10]. This relationship holds only in the high-SNR limit, which is also pivotal to the perceptual law. Previous work has shown that the allocation of Fisher information that maximizes mutual information between $x$ and $y$ is proportional to the square of the prior, such that

$$J(x) \propto p(x)^2. \tag{4}$$

The perceptual law of Wei & Stocker can be obtained by combining this formula with two other existing results relating Fisher information to bias and discriminability. First, Series, Stocker & Simoncelli 2009 [11] showed that Fisher information placed a bound on discriminability. In the high SNR regime, this bound can be made tight resulting in the identity, $D(x) \propto 1/\sqrt{J(x)}$, where the constant of proportionality depends on the desired threshold performance (e.g., 1 if the threshold $\delta \approx 76\%$). Second, the bias of a Bayesian ideal observer was shown in [8, 1] to relate to the prior distribution via the relationship $b(x) \propto \frac{d}{dx}\frac{1}{p(x)^2}$.

Combining these three proportionalities, we recover the perceptual law proposed by Wei & Stocker:

$$b(x) \overset{[1,8]}{\propto} \frac{d}{dx}\frac{1}{p(x)^2} \overset{[8]}{\propto} \frac{d}{dx}\frac{1}{J(x)} \overset{[11]}{\propto} \frac{d}{dx}D(x)^2. \tag{5}$$

Figure 2A illustrates the relationship between these quantities for a simulated example, highlighting its dependence on the high-SNR limit. In this paper, we will show that the condition $J(x) \propto p(x)^2$ is stronger than necessary, and that the same perceptual law arises from any allocation of Fisher information proportional to a power of the prior distribution, that is, $J(x) \propto p(x)^q$ for any $q > 0$.

Before showing this result, we first revisit the normative setting in which such power-law allocations of Fisher information are optimal.

## 3 Power-law efficient coding

We first show from where this power-law relationship between Fisher information and prior can emerge in an efficient neural code, and what factors determine the choice of power $q$. Previous work on information-maximizing or "infomax" codes [1, 8] has started from the following constrained optimization problem:

$$\underset{J(x)}{\arg\max} \int dx\, p(x) \log J(x) \quad \text{subject to } C(x) = \int dx\, \sqrt{J(x)} \leq c, \tag{6}$$

where $\log J(x)$ provides a well-known approximation to mutual information (up to an additive constant) as described above. Solving for the optimal Fisher information $J(x)$ using variational calculus and Lagrange multipliers produces (eq. 4) with the equality $J(x) = c^2\, p(x)^2$.

We can consider a more general method for defining normatively optimal codes by investigating Fisher information allocated according to

$$\underset{J(x)}{\arg\max} \, -\int dx\, p(x) J(x)^{-\alpha} \quad \text{subject to } C(x) = \int dx\, J(x)^{\beta} \leq c \tag{7}$$

with parameters $\alpha \geq 0$ defining the coding objective and $\beta > 0$ specifying a resource constraint. Several canonical normatively optimal coding frameworks emerge from specific settings of the parameter $\alpha$, independent of the value of $\beta$:

1. In the limit $\alpha \to 0$, this is equivalent to maximizing mutual information, since $\log J(x) = \lim_{\alpha \to 0} \frac{J(x)^{-\alpha} - 1}{-\alpha}$ [12].

2. If $\alpha = 1$, corresponds to minimizing the $L_2$ reconstruction error, sometimes called "discrimax" [6, 7] because it also optimizes squared discriminability.

3. For the the general case $\alpha = p/2$, for any $p > 0$, this optimization corresponds to minimizing the $L_p$ reconstruction error under the approximation $\mathbb{E}_{x,y}\left(|\hat{x} - x|^p\right) \approx \mathbb{E}_x\left(J(x)^{-p/2}\right)$, [12].

Here we show that this third relationship arises under a more general setting. We prove a novel bound on the mean $L_p$ error of *any* estimator for any level of SNR (see Supplemental Materials for proof, which builds on results from [13, 14]).

**Theorem** (Generalized Bayesian Cramer-Rao bound for $L_p$ error). *For any point estimator $\hat{x}$ of $x$, the mean $L_p$ error averaged over $x \sim p(x)$, $y|x \sim p(y|x)$, is bounded by*

$$\iint dx\,dy\, p(y, x)\Big|(\hat{x}(y) - x)\Big|^p \geq \int dx\, p(x) J(x)^{-p/2} \tag{8}$$

*for any $p > 0$, where $J(x)$ is the Fisher Information at $x$.*

Thus, the objective given in (eq. 7) captures a wide range of optimal neural codes via different settings of $\alpha$, including but not limited to classic efficient coding. We can solve this objective for any value of coding parameter $\alpha$ and constraint parameter $\beta > 0$ to obtain the optimal allocation of Fisher information. In all cases, the optimal Fisher information is proportional to the prior distribution raised to a power, which we therefore refer to as *power-law efficient codes*:

$$J_{\text{opt}}(x) = c^{1/\beta} \left( \frac{p(x)^{\gamma}}{\int dx\, p(x)^{\gamma}} \right)^{1/\beta} \triangleq k\, p(x)^q, \tag{9}$$

where $\gamma = \beta/(\beta + \alpha)$ and exponent $q = 1/(\beta + \alpha)$. (see Supplemental Materials for derivation). The normalized power function of the prior in parentheses is known as the escort distribution with parameter $\gamma$ [15]. Escort distributions arise naturally in power-law generalizations of logarithmic quantities such as mutual information, and could offer a reinterpretation of efficient coding and neural

coding more generally in terms of key theories such as maximum entropy, source coding, and Fisher information in generalized geometries [16, 17]. Here, we focus on the right-most expression, which characterizes a power-law efficient code in terms of the power $q$ and constant of proporationality $k = c^{1/\beta}(\int dx\, p(x)^{\gamma})^{-1/\beta}$. One interesting feature of the power-law efficient coding framework is that the exponent $q$, which determines how Fisher information is allocated relative to the prior, depends on both the coding parameter $\alpha$ and the constraint parameter $\beta$ via the relationship $q = 1/(\beta + \alpha)$. This implies that the optimal allocation of Fisher information is multiply determined, and reveals an ambiguity between coding desideratum and constraint in any optimal code.

In the particular case of infomax coding, where $\alpha = 0$, we obtain $q = 1/\beta$, meaning that the power law exponent $q$ is determined entirely by the constraint, and the escort parameter $\gamma$ equals 1. Previous work [7, 8, 12], therefore, could be interpreted to be implicitly or explicitly forcing the choice of $\beta = 1/2$. Any power-law efficient code with $J(x) = kp(x)^q$ could be putatively "infomax" if we defined the constraint such that $\beta = 1/q$. For example, the so-called discrimax encoder developed in [7] in which $J(x) \propto p(x)^{1/2}$ could result from an infomax objective function ($\alpha = 0$) if we only set the constraint $\beta = 2$. Rather than highlighting a pitfall of our procedure, this ambiguity instead highlights (*i*) the universality of the power-law generalization we present here, and (*ii*) the need to consider how other features of the observer model could further constrain the encoder to a uniquely infomax code.

## 4  Deriving linear and nonlinear bias-discriminability relationships

Next, we wish to go beyond proportionality and determine the precise relationship between bias and discriminability under the power-law efficient coding framework described above. However, any optimization of Fisher information, including ours, doesn't prescribe a method for selecting a parametric encoding distribution $p(y\,|\,x)$ associated with a particular power-law efficient code, that is, a distribution with Fisher information allocated according to $J(x) = k\,p(x)^q$. For simplicity, we therefore consider a power-law efficient code that is parametrized as Gaussian in $y$ with mean $x$:

$$p(y \mid x) = \mathcal{N}\Big(x, \frac{1}{kp(x)^q}\Big) = \sqrt{\frac{kp(x)^q}{2\pi}} \exp\Big(-\frac{kp(x)^q}{2}(y-x)^2\Big), \tag{10}$$

and we allocate Fisher information using a stimulus-dependent variance $\sigma^2 = 1/kp(x)^q$. This is the only configuration with that allocates Fisher information appropriately and is also is Gaussian in $y$. The parametrization of this encoder differs from that used by Wei and Stocker [1, 8], but critically has the same Fisher information. We can show that all key analytical results continue to hold in their parametrization when extended to power-law efficient codes, and that we ameliorate several issues in their models (see Supplemental Materials for comparisons and proofs). It also replicates the key results obtained with Wei and Stocker's parametrization, namely repulsive "anti-Bayesian" biases, in which the average Bayes least squares estimate is biased *away* from prior relative to the true stimulus [8, 18]. But we prefer this parametrization for its simplicity and interpretability in terms of its parameters $k$ and $q$.

At the decoding stage, Bayesian inference involves computing a posterior distribution over stimuli $x$, using the encoding distribution (eq. 10) as the likelihood:

$$p(x \mid y) = \frac{p(y \mid x)p(x)}{p(y)} = \frac{p(x)}{p(y)}\sqrt{\frac{kp(x)^q}{2\pi}} \exp\Big(-\frac{kp(x)^q}{2}(y-x)^2\Big). \tag{11}$$

In the high-SNR limit, the likelihood narrows and the $\log$-prior can be well-approximated with a quadratic about the true stimulus $x_0$, such that

$$\log p(x) \approx a_0 + a_1(x - x_0) + \tfrac{1}{2}a_2(x - x_0)^2$$

where the coefficients $a_0$, $a_1$, and $a_2$ are implicitly functions of $x_0$. For the MAP estimator $\hat{x}_{MAP}$, the bias in response to the stimulus at $x = x_0$ can be expressed in this limit as (see Supplemental Materials for proof)

$$b(x) = \frac{-\frac{(2+q)}{2q}\frac{1}{d_\delta'^2}\frac{d}{dx}D(x)^2}{1 - \Big(qa_1 - \frac{a_2}{a_1}\Big)\frac{(2+q)}{2q}\frac{1}{d_\delta'^2}\frac{d}{dx}D(x)^2}, \tag{12}$$

where $d'_\delta = \sqrt{2}Z(\delta)$ is the d-prime statistic for a fixed performance $\delta$, and $Z(\cdot)$ is the inverse normal CDF. We refer to this as our *nonlinear relationship* because it expresses bias $b(x)$ as a nonlinear function of the squared discriminability $D(x)^2$. This relationship makes testable nonlinear predictions between bias and discriminability that depend on the shape of the prior at each value of $x$ through the local prior curvature parameters $a_1$ and $a_2$.

We recover a linear relationship between bias and discriminability in the higher-SNR limit when $|\frac{d}{dx}D(x)^2| \ll |\frac{(2+q)}{2q}\frac{1}{d'^2_\delta}(qa_1 - \frac{a_2}{a_1})|^{-1}$, satisfied if the SNR $k \gg |\frac{(2+q)}{2q}(qa_1 - \frac{a_2}{a_1})e^{-qa_0}|$ for all $x_0$. This specification of the high-SNR regime reveals that the likelihood must be so sharp around the stimulus that the prior, by comparison, becomes so broad that it is nearly flat. When satisfied, the final result is the following *linear relationship* between bias and discriminability:

$$b(x) = \left( -\frac{(2+q)}{2q}\frac{1}{d'^2_\delta} \right) \frac{d}{dx}D(x)^2, \tag{13}$$

which indicates a negative constant of proporationality for all $q$. There is no contribution of $a_1$ or $a_2$ to the coefficient of proportionality; only $q$ matters. Thus, we confirm that for power-law efficient codes generally, the Wei-Stocker law $b(x) \propto \frac{d}{dx}D(x)^2$ holds in the limit of high SNR for all $x$.

## 5    Simulating the model under different SNRs and power-law efficient codes

We used simulated data to test our derived nonlinear and linear relationships between bias and discriminability (eqs. 12 & 13). We restricted these simulations to the high-SNR regimes in which the analytical predictions provide accurate descriptions of the simulated data, and examined the qualitative differences that emerge for different powers of the power-law efficient code. We consider a sweep of both of these parameters, $k$ and $q$, under different decoder loss functions, which yield different Bayesian estimators with very different implications for the resulting bias.

In all simulations, we propagate each stimulus $x \sim p(x)$ on a finely tiled grid through a Bayesian observer model numerically, computing a posterior $p(x|y) \propto p(x)\mathcal{N}(y; x, kp(x)^{-q})$ for a power-law efficient code under many powers $q$ and SNRs $k$, and for each computed the Bayesian estimators associated with various loss functions of interest. We repeated this procedure for a large number of random smooth priors. The bias-discriminability relationship will be most clearly observed if our data can tile the space of discriminability and bias, achieved if the underlying priors are maximally diverse and rich in curvature. As such, we draw random priors as exponentiated draws from Gaussian processes on $[-\pi, \pi]$, according to

$$p(x) = \tfrac{1}{Z}\exp(\mathbf{f}), \quad \text{where } \mathbf{f} \sim \mathcal{GP}(0, K) \tag{14}$$

for $Z$ as a normalizing constant, and $K$ the radial basis function kernel wherein

$$K_{ij} = \rho\exp\left(\tfrac{1}{2\ell^2}\|x_i - x_j\|^2\right) \tag{15}$$

with magnitude $\rho = 1$ and lengthscale $\ell = 0.75$, selected such that a typical prior was roughly bimodal. In this way, the vector elements are artifically ordered on a line to enforce smoothness. To prevent truncating probability mass at the endpoints of the domain, we only record measurements on the interior subinterval $[-\pi/2, \pi/2]$.

While we offer more details in the following sections, we first overview briefly the goals of the two remaining figures. In Figure 4, we explore how quality of predictions made by the nonlinear and linear relationships in (eqs. 12 and 13) change as a function of the SNR $k$ for various power-law efficient coding powers $q$. In Figure 5, we observe how the slope of the [linear] relationship changes as a function of $q$, to which we can compare our analytical predictions to simulated results.

### 5.1    Tests of prior-dependent nonlinear and prior-*independent* linear relationships

The nonlinear and linear bias-discriminability relationships together form a broad generalization of the perceptual law beyond Wei and Stocker's prior work [1]. As SNR increases and the relationship converges onto a line (Figure 2D), the fluctuations along that line are captured by both relationships, but the nonlinear relationship captures some additional fluctuations orthogonal to the predicted line (Figure 3). Both nonlinear and linear relationships are exceptional approximations of the true bias

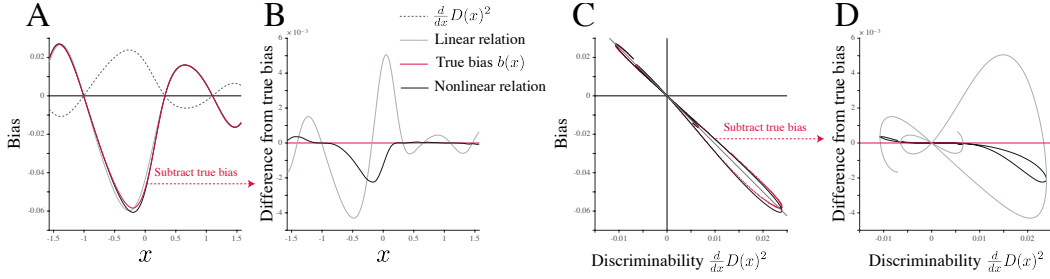

Figure 3: Nonlinear and linear bias-discriminability relationships for SNR $k = 10^2$ and "discrimax" code $q = 1/2$ under an exemplar random prior. Bias and discriminability match closely under the linear relationship (A), but any deviations from that line are well-captured by the weak nonlinear relationship (C). Deviations from the true bias (red) are best observed if we subtract the true bias from the predictions of the linear and nonlinear models (gray and black curves, respectively; B, D).

(Figure 3A), but do not capture equivalent features of the curvature – deviations are often at very different values of $x$ (Figure 3B). We can equivalently view this parametrically as a function of discriminability (Figure 3C, D).

We quantify the quality of our nonlinear and linear predictions as a function of SNR by measuring an error ratio $R$, defined as the ratio between the mean-squared error of the bias predictions under a model (nonlinear, linear) and the total mean-squared error, such that

$$R = -\log\left(\frac{MSE_{\text{model}}}{MSE_{\text{null}}}\right) = -\log\left(\frac{\int dx \, (\hat{b}_{\text{model}} - b(x))^2}{\int dx \, b(x)^2}\right) \tag{16}$$

where $\hat{b}_{\text{model}}$, for clarity, represents a bias predicted under a given relationship (eqs. 12 or 13). We use the negative logarithm such that $R > 0$ imply model predictive performance better than null. This ratio is defined for each prior, which we then average over 200 random priors for all simulations.

The null model in all cases is 0 everywhere. We want each simulation's mean-squared error to be normalized according to how much bias the underlying prior introduced – if the prior were flat, our Gaussian encoding model is unbiased and symmetric for all moments such that bias is 0 everywhere. For MAP estimation, we use our analytical nonlinear and linear relationships as the models (Figure 4A,B), further using the difference between the two $\Delta R = -\log(MSE_{\text{nonlin}}/MSE_{\text{lin}})$ to measure the relative performance of each model to the other (Figure 4C). We only highlight the regions where both models are making sensible predictions ($R > 0$). For posterior median and mean computation, in the absence of analytical results, we use as the model a linear function regressed to the data. While by definition the estimated $R > 0$, the degree to which it's positive makes it still a useful surrogate for measuring the relative linearity (Figure 4D,E).

The bias-discriminability relationship emerges from modest SNR $k$ for any estimator (MAP, posterior median, posterior mean) and power-law efficient code with power $q$, converging into the linear relationship as SNR $k$ increases (Figure 4). The analytical results for the MAP estimator model the data well, as the linear and nonlinear error ratio measures cleanly cross 0 and peak (Figure 4A,B). The decrease after this peak is a numerical precision issue and isn't informative of perceptual processing – both bias and discriminability measurements collapse into zero as $k$ increases. The minimum SNR required for good predictions is lower for the nonlinear relationship, and this form makes better predictions than the linear relationship throughout, evidenced by the error ratio difference $\Delta R$ being positive (Figure 4C). Moreover, the slope of the relationship as predicted from (eq. 13), as a function of $q$, exactly matches simulations (Figure 5A).

## 5.2 Posterior median and mean estimators, anti-Bayesian repulsive biases

Analytical results for posterior median and posterior mean estimators are nontrivial, and beyond the scope of this work. However, they are likely tractable, and simulations offer interesting insight into potentially useful functional forms of an equivalent linear bias-discriminability relationship in the high-SNR limit. The posterior median could be asymptotically unbiased in $q$ or unbiased at $q = 2$, as the bias tends to 0 rapidly, and the linear relationship erodes (Figure 4D, Figure 5B). The posterior

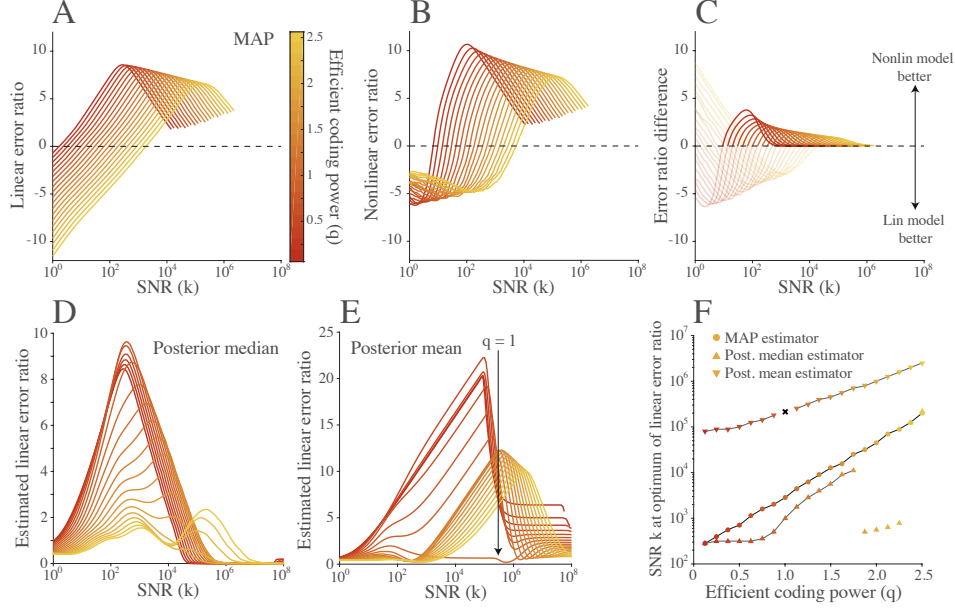

Figure 4: Linearity and nonlinearity indices of analytical predictions (MAP, A-C) or regression fits (Posterior median and mean, D-E) as $k$ increases. A, B. error ratio $R$ of linear and nonlinear relationships, respectively, as a function of increasing SNR $k$ and increasing efficient coding power $q$ as the color brightens from red to yellow. C. Error ratio difference $\Delta R$ shows a lower minimal SNR for the nonlinear model to make effective predictions of bias than the linear model. Regions in which either model is not making sensible predictions ($R < 0$) are faded. D, E. Estimated linear error ratio (by regression) for posterior median and mean estimators, respectively. F. Optimal SNR for linear bias-discriminability as a function of efficient coding power and Bayesian estimator.

mean, on the other hand, is asymptotically unbiased for $q = 1$ and has repulsive biases away from the prior for $q > 1$ (Figure 4E), a hallmark of the Bayesian observer introduced by Wei and Stocker previously [8]. Although we have not developed a formal derivation, we propose the following simple relationship parametrizing the slope, after using curve-fitting to explore various functional forms:

$$b(x) \overset{?}{=} \frac{\log(q)}{\sqrt{q}} \frac{1}{d_\delta'^2} \frac{d}{dx} D(x)^2 \tag{17}$$

$q = 1$ is a natural transition point for these attractive-repulsive biases (see the zero-crossing in Figure 5C). Recalling (13), in this setting, the Fisher information is simply a scaling of the prior. For $q < 1$, low-probability events have boosted probability mass since $p(x) < p(x)^q$. Meanwhile, for $q > 1$, these same events have compressed probability mass since $p(x) > p(x)^q$. For a power-law efficient code, $q$ is determining the weight of the tails of this likelihood. In this way, the specific infomax setting of $q = 2$ demonstrates repulsive biases not because it corresponds to a mutual information-maximizing encoder, but because of the tail behaviors it induces by being greater than 1.

## 6 Discussion

We have shown that a perceptual law governing the relationship between perceptual bias and discriminability arises under a wide range of Bayesian optimal encoding models. This extends previous work showing that the law arises from information-theoretically optimal codes [1], which our work includes as a special case. Maximization of mutual information therefore does not provide a privileged explanation for the neural codes underlying human perceptual behavior, in the sense that the same lawful relationship emerges for all members of the more general family of power-law efficient codes. We have also extended the perceptual law put forth by Wei and Stocker by deriving the exact constant of proportionality between bias and derivative of squared discriminability for arbitrary choices of power-law exponent.

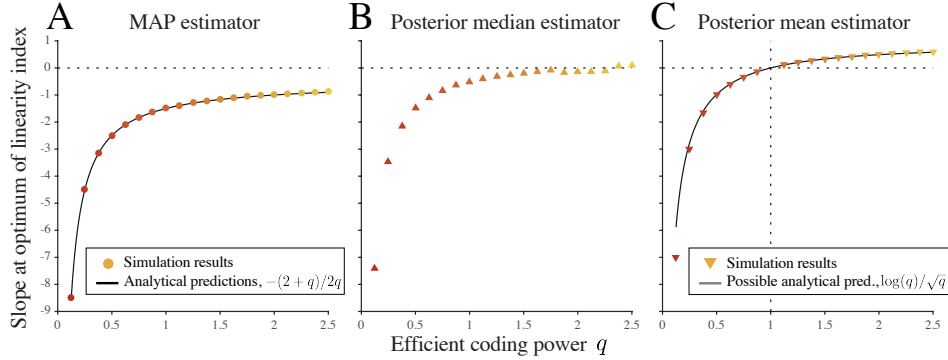

Figure 5: Linear slope of the bias-discriminability relation as a function of the efficient coding power $q$. A. MAP estimator for analytical predictions (solid line) and simulations (dots). B. Posterior median estimator for simulations. C. Posterior mean estimator for simulations fit parsimoniously by a simple equation. Note that the slope changes sign after $q = 1$ (vertical line). Before this crossing, biases are prior-attractive ($q < 1$), and after are prior-repulsive, or "anti-Bayesian" ($q > 1$).

More generally, we have shown that power-law efficient codes arise under a general optimization program that trades off the cost of making errors against a constraint on the total Fisher information (eq. 7). Any particular allocation of Fisher information relative to the prior is therefore optimal under multiple settings of loss function and constraint, and information-theoretically optimal coding is consistent with a range of different power-law relationships between Fisher information and prior. This implies that the form of an optimal power-law efficient code depends on specifying a choice of constraint as well as a choice of loss function.

Although our work shows that Wei and Stocker's perceptual law is equally consistent with multiple forms of optimal encoding, other recent work has suggested that information-maximization provides a better explanation of both perceptual and neural data than other loss functions [19]. One interesting direction for future work will be to determine whether other members of the power-law efficient coding family can provide equally accurate accounts of such data.

Another direction for future work will be to consider more general families of efficient neural codes. We hypothesize that, since power functions form a basis set for *any* function, we could show that Wei and Stocker's law emerges whenever neural resources are allocated according to any strictly monotonic function of the prior (with positive support). Such an efficient coding principle could imply

$$J(x) \propto G\big(p(x)\big) \overset{?}{\Longrightarrow} b(x) \propto \frac{d}{dx} D(x)^2 \quad \text{for strictly monotone } G : \{p(x) \mid x \in \mathcal{X}\} \to \mathbb{R}^+ \quad (18)$$

Critically, various specialized neural circuits throughout the brain needn't adopt the same power-law $q$, or function $G(\cdot)$. The end result is the same: biases nudge perceptual estimates towards stimuli that are more (or potentially less) discriminable (confer eq. 1, bias is a scaled step along the gradient of discriminability). Neural populations could therefore specialize computations by refining $q$ or $G(\cdot)$ to precisely privilege or discount representations of stimuli with different prior probabilities. Mutual information is one of many such specializations, and is likely sensible under some conditions, but not necessarily all. In this way, the bias-discriminability relationship could be the signature of a unifying organizational principle governing otherwise diverse neural populations that encode sensory information. It could be useful to reconceptualize "efficient codes" accordingly as a broad family of codes governed by this more general normative principle, within which an efficient code putatively allocates neural resources such that stimuli that are common under the prior are encoded more faithfully than stimuli that are uncommon under the prior. We note that this echoes our initial intuitions of a good encoder, and we've provided evidence to suggest that this simple condition could be sufficient.

### Acknowledgments

We thank David Zoltowski and Nicholas Roy for helpful comments. MJM was supported by an NSF Graduate Research Fellowship; JWP was supported by grants from the McKnight Foundation, Simons Collaboration on the Global Brain (SCGB AWD1004351) and NSF CAREER Award (IIS-1150186).

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
