[Supplementary Material]

# Supplementary Materials

## 1  Variational calculus derivation of power-law efficient coding

Optimizing (eq. 9) in the main text follows from a straightforward variational calculus optimization using Lagrange multipliers. Defining the functional as $\mathcal{I}$, we can setup the optimization in (eq. 9) as

$$\mathcal{I} = -\int dx\, p(x)J(x)^{-\alpha} + \lambda\left(\int dx\, J(x)^{\beta} - c\right) \tag{19}$$

for $\alpha > 0$ and $\beta > 0$ (recall that the specific case of infomax is $\alpha \to 0$). Calculating the optimal Fisher information $J(x)$ such that $\nabla\mathcal{I} = 0$, we find

$$\frac{\partial\mathcal{I}}{\partial J(x)} = 0 = \alpha p(x)J'(x)J(x)^{-\alpha-1} + \beta\lambda J'(x)J(x)^{\beta-1}$$

$$\implies 0 = J'(x)\Big[\alpha p(x)J(x)^{-\alpha-1} + \beta\lambda J(x)^{\beta-1}\Big]$$

We assert that $J'(x) \neq 0$ since we desire a solution that depends on environmental statistics. That leaves the second term equalling zero, from which follows that

$$-\alpha p(x)J(x)^{-\alpha-1} = \beta\lambda J(x)^{\beta-1}$$

$$\implies J(x) = \left(\frac{-\alpha}{\lambda\beta}\, p(x)\right)^{1/(\beta+\alpha)} \tag{20}$$

We solve for the Lagrange multiplier $\lambda$ using the other partial derivative:

$$\frac{\partial\mathcal{I}}{\partial\lambda} = 0 \implies \int dx\, J(x)^{\beta} = c$$

$$\text{substituting (20)} \implies \left(\frac{-\alpha}{\beta\lambda}\right)^{\beta/(\beta+\alpha)} \int dx\, p(x)^{\beta/(\beta+\alpha)} = c$$

$$\implies \lambda = -\frac{\alpha}{\beta}\frac{1}{c^{(\beta-\alpha)/\beta}}\left(\int dx\, p(x)^{\beta/(\beta+\alpha)}\right)^{(\beta+\alpha)/\beta}$$

If we define a surrogate variable $\gamma = \beta/(\beta+\alpha)$, we can redefine

$$\lambda = -\frac{\alpha}{\beta}\frac{1}{c^{1/\gamma}}\left(\int dx\, p(x)^{\gamma}\right)^{1/\gamma} \tag{21}$$

$$\implies J(x) = \left(\frac{\alpha}{\beta}\frac{\beta c^{1/\gamma}}{\alpha}\frac{p(x)}{\left(\int dx\, p(x)^{\gamma}\right)^{1/\gamma}}\right)^{1/(\beta+\alpha)}$$

$$\implies J(x) = c^{1/\beta}\left(\frac{p(x)^{\gamma}}{\int dx\, p(x)^{\gamma}}\right)^{1/\beta} \longrightarrow kp(x)^{q} \tag{22}$$

if $k \triangleq c^{1/\beta}(\int dx\, p(x)^{\gamma})^{-1/\beta}$ and $q \triangleq 1/(\beta+\alpha)$, where we substitute exponents with some algebra for clarity, and report this result as equation (9) in the main text.

### 1.1  Generalized Bayesian Cramer-Rao bound for $L_2$ and $L_p$ errors

The proof of our generalized Bayesian Cramer-Rao bound, a lower bound on the mean-$L_p$ error of an estimator, follows from a hybrid of those of the van Trees inequality [13] and the Barakin-Vajda Cramer-Rao inequality [20], redeveloped for our purposes. To begin, the bias of *any* estimator $\hat{x}(y)$ of a stimulus $x$ is given by

$$b(x) = \int dy\, p(y \mid x)\big(\hat{x}(y) - x\big)$$

Multiplying the prior to each side and differentiating with respect to $x$, we get

$$\frac{\partial}{\partial x}p(x)b(x) = \frac{\partial}{\partial x}\int dy\, p(y\mid x)p(x)\big(\hat{x}(y)-x\big)$$

$$= -\int dy\, p(y\mid x)p(x) + \int dy\, \frac{\partial}{\partial x}\Big(p(y\mid x)p(x)\Big)\big(\hat{x}(y)-x\big)$$

Now integrating with respect to $x$ and substituting the joint distribution $p(y,x)$ where appropriate,

$$p(x)b(x)\Big|_{\mathcal{X}} = -\iint dxdy\, p(y,x) + \iint dxdy\, \Big(\frac{\partial}{\partial x}p(y,x)\Big)\big(\hat{x}(y)-x\big)$$

$$\implies 0 = -1 + \iint dxdy\, p(y,x)\Big(\frac{\partial}{\partial x}\log p(y,x)\Big)\big(\hat{x}(y)-x\big)$$

where the zero holds if $b(x)=0$ or $p(x)=0$ at the endpoints of the domain of $x$ (or the two are equal at those endpoints). We can now apply Hölder's inequality to the integral, a generalization of the Cauchy-Schwarz inequality stating that, for Hölder conjugates $p, u > 1$ such that $p^{-1}+u^{-1}=1$, and functions $f(y,x)$ and $g(y,x)$, we have

$$\left|\iint dxdy\, p(y,x)f(y,x)g(y,x)\right| \le \iint dxdy\, p(y,x)\big|f(y,x)g(y,x)\big|$$

$$\le \left(\iint dxdy\, p(y,x)\big|f(y,x)\big|^{p}\right)^{1/p}\left(\iint dxdy\, p(y,x)\big|g(y,x)\big|^{u}\right)^{1/u}$$

Set $f(y,x) = (\hat{x}(y)-x)$ and $g(y,x) = \frac{\partial}{\partial x}\log p(y,x)$ such that

$$1 \le \left(\iint dxdy\, p(y,x)\Big|\frac{\partial}{\partial x}\log p(y,x)\Big|^{u}\right)^{1/u}\left(\iint dxdy\, p(y,x)\big|(\hat{x}(y)-x)\big|^{p}\right)^{1/p}$$

Exponentiating everything to the $p$-th power and rearranging, we see that

$$\iint dxdy\, p(y,x)\big|(\hat{x}(y)-x)\big|^{p} \ge \left(\iint dxdy\, p(y,x)\Big|\frac{\partial}{\partial x}\log p(y,x)\Big|^{u}\right)^{-p/u} \tag{23}$$

$$\ge \left(\int dx\, p(x)\left(\int dy\, p(y\mid x)\Big|\frac{\partial}{\partial x}\log p(y,x)\Big|^{2}\right)^{u/2}\right)^{-p/u}$$

$$\ge \left(\int dx\, p(x)\left(\int dy\, p(y\mid x)\Big|\frac{\partial}{\partial x}\log p(y,x)\Big|^{2}\right)^{p/2}\right)^{-1}$$

$$= \left(\int dx\, p(x)\left(-\int dy\, p(y\mid x)\left[\frac{\partial^{2}}{\partial x^{2}}\log p(y\mid x) + \frac{\partial^{2}}{\partial x^{2}}\log p(x)\right]\right)^{p/2}\right)^{-1}$$

after applying Jensen's inequality twice. We note that equation (23) is a generalization of the Bayesian Cramer-Rao inequality, sometimes called the van Trees inequality after [13], to the mean-$L_p$ error of an estimator $\hat{x}$.

We can make several substitutions: the left-hand side is the mean-$L_p$ error of the estimator $\hat{x}$, and the right-hand side contains two partial derivatives – the first is the Fisher information, and the second is the curvature of the prior, a constant with respect to the efficient coding optimization problem. As Fisher information increases in the high-SNR limit, this term becomes negligible [14], and we see that

$$\mathbb{E}_{x,y}\Big(|\hat{x}(y)-x|^{p}\Big) \ge \left(\mathbb{E}_{x}\left(\left[J(x)-\frac{\partial^{2}}{\partial x^{2}}\log p(x)\right]^{p/2}\right)\right)^{-1} \longrightarrow \left(\mathbb{E}_{x}\Big(J(x)^{p/2}\Big)\right)^{-1}$$

and $\quad -\mathbb{E}_{x,y}\Big(|\hat{x}(y)-x|^{p}\Big) \le -\mathbb{E}_{x}\Big(J(x)^{-p/2}\Big) \tag{24}$

with equality under a chain of conditions, all satisfied when Hölder's inequality holds with equality, occuring when, for a constant $A$,

$$\frac{\partial}{\partial x}\log p(y,x) = A(\hat{x}(y) - x) \implies \frac{\partial^2}{\partial x^2}\log p(y,x) \equiv \frac{\partial^2}{\partial x^2}\log p(x \mid y) = -A$$
$$\implies p(x \mid y) = \exp(-Ax^2 + c_1 x + c_2)$$

Equivalently, the bound is tight as the posterior becomes Gaussian, satisfied in the high-SNR limit we consider throughout. We use the negative of equation (24) to state the theorem in the main text.

## 2 Full derivation of nonlinear and linear bias-discriminability relations

We complete the derivation of the bias-discriminability relations in detail, in an attempt to highlight why each assumption or approximation is necessary to move the derivation forward. After some algebra pulling the functions of the prior into the exponent, we can equivalently express the posterior in (15) as

$$p(x \mid y) = \frac{1}{p(y)}\sqrt{\frac{k}{2\pi}}\exp\left(k\left[-\frac{p(x)^q}{2}(y-x)^2 + \left(\frac{2+q}{2q}\right)\frac{q}{k}\log p(x)\right]\right) \tag{25}$$

The maximum a posteriori estimator $\hat{x}_{MAP} \triangleq x^*$ will have a bias measurable from the argument of the exponential in the square brackets. The maximum of the log-posterior is that derivative:

$$\left.\frac{\partial \log p(x \mid y)}{\partial x}\right|_{x=x^*} = 0 = -\frac{q}{2}p'(x^*)p(x^*)^{q-1}(y-x^*)^2 + p(x^*)^q(y-x^*) + \left(\frac{2+q}{2q}\right)\frac{q}{k}\frac{p'(x^*)}{p(x^*)} \tag{26}$$

and when we solve this quadratic equation for $y - x^*$,

$$y - x^* = \frac{-1}{q\,p'(x^*)p(x^*)^{q-1}}\left(-p(x^*)^q \pm \sqrt{p(x^*)^{2q} + (2+q)\frac{q}{k}\frac{p'(x^*)^2}{p(x^*)}p(x^*)^{q-1}}\right)$$
$$= \frac{p(x^*)}{q\,p'(x^*)}\left(1 \mp \sqrt{1 + (2+q)\frac{q}{k}\frac{p'(x^*)^2}{p(x^*)^{q+2}}}\right)$$

and we can simplify using the binomial expansion, $(1+x)^\alpha \approx 1 + \alpha x$ when $|\alpha x| \ll 1$ if we truncate terms of order $1/k^2$. Only the negative root that cancels the leading 1 is reasonable, leaving

$$y - x^* = \frac{p(x^*)}{q\,p'(x^*)}\left(1 \mp \left[1 + \left(\frac{2+q}{2}\right)\frac{q}{k}\frac{p'(x^*)^2}{p(x^*)^{q+2}} + \mathcal{O}\left(\frac{1}{k^2}\right)\right]\right) \longrightarrow -\left(\frac{2+q}{2q}\right)\frac{q}{k}\frac{p'(x^*)}{p(x^*)}\frac{1}{p(x^*)^q}$$
$$\text{when } k \gg \max_x \left(\frac{2+q}{2q}\right)\left(\frac{q\,p'(x)}{p(x)}\right)^2\frac{1}{p(x)^q}$$

the conditions on which can be satisfied in the high-SNR limit by setting $k$ sufficiently large as dictated here. Indeed, this inequality suggests a benchmark for measuring when the high-SNR limit begins to emerge in the model. We note that this is equivalent to truncating the quadratic term of (25) outright, but our derivation clarifies this truncation as a function of the prior and SNR.

As the SNR $k$ increases, the likelihood sharpens about the true stimulus $x_0$, such that the prior by comparison dilates and could be well-approximated locally by a log-linear or quadratic function about $x_0$. We consider the latter for generality, and find that

$$\log p(x^*) \approx a_0 + a_1(x^* - x_0) + \frac{1}{2}a_2(x^* - x_0)^2$$
$$\text{and } p(x^*)^q = \exp(qa_0)\exp\left(q\left[a_1(x^* - x_0) + \frac{1}{2}a_2(x^* - x_0)^2\right]\right) \triangleq p(x_0)^q/\epsilon_{x_0}(y)$$

where we define $\epsilon_{x_0}(y) = \exp\left(-q[a_1(x^* - x_0) + \frac{1}{2}a_2(x^* - x_0)^2]\right)$ as the residual error in approximating $p(x^*)$ as $p(x_0)$. Since $a_1 + a_2(x^* - x_0) \equiv p'(x^*)/p(x^*)$ and $x^* - x_0$ will always

have opposite signs, we can enforce not only that $\epsilon_0(y) \in (0,1]$, but also that $\epsilon_0(y) \longrightarrow 1$ as the SNR increases in our asymptotic regime. We can solve for the bias readily, substituting these redefinitions of the prior:

$$y - x^* = (y - x_0) - (x^* - x_0) = -\frac{(2+q)}{2q}\frac{qa_1}{kp(x_0)^q}\epsilon_{x_0}(y)\Big[1 + \frac{a_2}{a_1}(x^* - x_0)\Big]$$

$$\implies b(x_0) = \mathbb{E}_{y|x_0}\big(x^* - x_0\big) = \mathbb{E}_{y|x_0}\big(y - x_0\big) + \frac{(2+q)}{2q}\frac{qa_1}{kp(x_0)^q} \cdot \mathbb{E}_{y|x_0}\Big(\epsilon_{x_0}(y)\Big[1 + \frac{a_2}{a_1}(x^* - x_0)\Big]\Big)$$

and $\mathbb{E}_{y|x_0}(y - x_0) = 0$ will follow as the first central moment of the Gaussian encoding distribution. Expanding the residual and again truncating $\mathcal{O}(1/k^2)$ terms, we can resolve a more elaborate expression for the bias:

$$b(x_0) = \Big(\frac{2+q}{2q}\Big)\frac{qa_1}{kp(x_0)^q}\left[1 + \Big(\frac{a_2}{a_1} - qa_1\Big)b(x_0)\right]$$

We leverage prior results [11] to state the proportionality in (5) with equality, and the derivative of this relation is given by

$$D(x_0)^2 = \frac{d_\delta'^2}{J(x_0)} = \frac{d_\delta'^2}{kp(x_0)^q} \implies -\frac{\frac{d}{dx}D(x_0)^2}{d_\delta'^2} = \frac{qp'(x_0)}{p(x_0)}\frac{1}{kp(x_0)^q} = \frac{qa_1}{kp(x_0)^q}$$

We then solve for the bias:

$$b(x_0) = \frac{-\frac{(2+q)}{2q}\frac{1}{d_\delta'^2}\frac{d}{dx_0}D(x_0)^2}{1 - \Big(qa_1 - \frac{a_2}{a_1}\Big)\frac{(2+q)}{2q}\frac{1}{d_\delta'^2}\frac{d}{dx_0}D(x_0)^2} \tag{27}$$

and report this as equation (12) in the main text.

## 3 Replication with Wei and Stocker's alternative encoding model

However, Wei and Stocker do not build up their observer model under these noise assumptions. We frame the encoding problem in the high-SNR limit as a noise allocation problem, optimizing the Fisher information with low-noise priority for more frequently-observed stimuli. They frame the encoding problem in this limit as a stimulus remapping problem, optimizing a monotonic warping function $f(\cdot)$ to further separate adjacent stimuli in a "sensory space" with the same prior-driven prioritization. We define their observer model as having $f$-normally distributed noise since the sensory-space representation $\tilde{y} = f(y)$ is normally distributed, by analogy to the log-normal distribution. Under $f$-normal noise assumptions, they posit a likelihood of the form

$$p(y \mid x) = \mathcal{N}(f(y); \; f(x), 1/k) = f'(y)\sqrt{\frac{k}{2\pi}}\exp\Big(-\frac{k}{2}\big(f(y) - f(x)\big)^2\Big) \tag{28}$$

where the $\sigma^2$ noise term in their framework is equivalent to our SNR term $1/k$, which emerges implicitly in $f$, since the Fisher information would become a simple function of $f(\cdot)$ and $k$,

$$J(x) = \frac{f'(x)^2}{\sigma^2} \mapsto kf'(x)^2 \tag{29}$$

Recalling our power-law efficient coding hypothesis, which stated $J(x) = kp(x)^q$, we could solve for $f(\cdot)$ and restate the constraints applied by Wei and Stocker,

$$f(x) = \frac{1}{\sqrt{k}}\int_{-\infty}^{x} dx'\sqrt{J(x')} = \int_{-\infty}^{x} dx' \, p(x')^{q/2} \leq c \tag{30}$$

This constraint would not be particular to infomax efficient coding when stated in terms of the Fisher information, and would merely require that the support of $f(\cdot)$ be finite, or biologically speaking, that the neural resources allocated by the encoder are limited.

However, their $f$-normal model, as they present it, would not be a statistically viable alternative model. Namely, the likelihood isn't a proper distribution, and doesn't integrate to 1 due to the finite support of the warping function $f(\cdot)$, so restricted by their finite-resource constraint. If we integrate the encoding distribution $p(y \mid x)$ with respect to $y$, we observe that, due to the resource constraint,

$$\int dy\, p(y \mid x) = \Phi\big[\sqrt{k}(c - f(x))\big] - \Phi\big[-\sqrt{k}f(x)\big]$$

where we use $\Phi[\cdot]$ to denote the cumulative standard normal distribution. The proper encoding distribution would then take on a form like

$$p(y \mid x) \approx \frac{2f'(y)}{\Phi\big[\sqrt{k}(c - f(x))\big] - \Phi\big[-\sqrt{k}f(x)\big]} \sqrt{\frac{k}{2\pi}} \exp\Big(-\frac{k}{2}\big(f(y) - f(x)\big)^2\Big) \qquad (31)$$

While still semblant of a Gaussian, the posterior loses analytical tractability; moreover, the Fisher information and therefore the warping function $f(\cdot)$ would have changed from the original forms since we added a function of $x$, and we cannot compute either as a result. As the SNR $k \to \infty$, the first $\Phi$ function tends to 1 while the second tends to zero, recovering their original form. But it wouldn't be desirable to select for an encoding model that's *only* viable in the high-SNR regime, since any comparison to that model's behavior in a low-SNR state is ill-posed. It may be unclear whether the emergent statistical properties are due to the transition into the high-SNR regime or the transition into well-defined encoding distributions, among other potential pitfalls.

For these reasons, we elected to propose the Gaussian encoding model, noting it is the only encoding distribution that could be Gaussian in $y$ satisfying the Fisher information constraints, and it remains analytically tractable and interpretable in its parameters. That said, if we eschew concerns with the viability of Wei and Stocker's model, it is straightforward to replicate our core results of power-law efficient coding and the bias-discriminability proportionality within their framework.

### 3.1 Generalizing biases of Wei and Stocker's model for power-law efficient codes

Since the definition of discriminability follows only from the high-SNR limit through the Cramer-Rao bound, and does not depend on efficient coding, they are the same under Normal and $f$-normal models. We need only replicate the expressions they derive for the biases of Bayesian estimators under various loss functions. We follow their proofs line-by-line as allowed, and adopt their prime-notation $(\cdot)'$ for derivatives for more immediate comparability. Note that at any stage, we could substitute $q = 2$ and $1/k = \sigma^2$ and retrieve their results precisely. The posterior median derivation, hinging on the Taylor expansion of $f^{-1}(\cdot)$, becomes a natural building block for all other analytical results they present. We present this expansion equivalently in terms of the warping function $f(\cdot)$, as prematurely substituting for the prior occludes the generality of the result. The relationship between the bias $b(x_0)$ and the derivative of the inverse prior-squared $\big(1/p(x)^2\big)'_{x_0}$, as they present it, is only meaningful insofar as that prior-squared term is equal to the Fisher information. The Fisher information is ultimately what connects the bias to the discriminability, whether the power is 2 or a more general $q$.

$L_1$ **norm-minimizing posterior median estimator** Wei and Stocker observed that in their framework, the posterior median $\bar{x} = y$, such that the bias under this estimator becomes

$$b(x_0) = \mathbb{E}_{y|x_0}(\bar{x} - x_0) = \mathbb{E}_{y|x_0}(y) - x_0$$
$$= \sqrt{\frac{k}{2\pi}} \int dy\, y \cdot f'(y) \exp\Big(-\frac{k}{2}\big(f(y) - f(x_0)\big)^2\Big) - x_0$$
$$= \sqrt{\frac{k}{2\pi}} \int d\tilde{y}\, f^{-1}(\tilde{y}) \exp\Big(-\frac{k}{2}(\tilde{y} - \tilde{x}_0)^2\Big) - f^{-1}(\tilde{x}_0)$$

Using the same second-order Taylor expansion of $f^{-1}(\cdot)$, it follows that

$$f^{-1}(\tilde{y}) \approx f^{-1}(\tilde{x}_0) + \big(f^{-1}(\tilde{y})\big)'_{\tilde{y}=\tilde{x}_0}(\tilde{y} - \tilde{x}_0) + \frac{1}{2}\big(f^{-1}(\tilde{y})\big)''_{\tilde{y}=\tilde{x}_0}(\tilde{y} - \tilde{x}_0)^2$$

where the derivative is with respect to $\tilde{y}$ evaluated at $\tilde{x}_0$. Applying the expansion into the integral equation,

$$\mathbb{E}_{y|x_0}(y) - x_0 = \sqrt{\frac{k}{2\pi}} \int d\tilde{y} \, \frac{1}{2} \left( f^{-1}(\tilde{y}) \right)''_{\tilde{y}=\tilde{x}_0} (\tilde{y} - \tilde{x}_0)^2 \exp \left( -\frac{k}{2}(\tilde{y} - \tilde{x}_0)^2 \right)$$

$$= \sqrt{\frac{k}{2\pi}} \int d\tilde{y} \, \frac{1}{2} \left( \frac{1}{f'(f^{-1}(\tilde{y}))} \right)'_{\tilde{y}=\tilde{x}_0} (\tilde{y} - \tilde{x}_0)^2 \exp \left( -\frac{k}{2}(\tilde{y} - \tilde{x}_0)^2 \right)$$

$$= \frac{1}{2} \sqrt{\frac{k}{2\pi}} \int d\tilde{y} \left( -\frac{f''(x_0)}{f'(x_0)^3} \right) (\tilde{y} - \tilde{x}_0)^2 \exp \left( -\frac{k}{2}(\tilde{y} - \tilde{x}_0)^2 \right)$$

$$= \frac{1}{4} \sqrt{\frac{k}{2\pi}} \int d\tilde{y} \left( \frac{1}{f'(x_0)^2} \right)'_{x_0} (\tilde{y} - \tilde{x}_0)^2 \exp \left( -\frac{k}{2}(\tilde{y} - \tilde{x}_0)^2 \right)$$

$$= \frac{1}{4} \left( \frac{1}{f'(x_0)^2} \right)'_{x_0} \cdot \sqrt{\frac{k}{2\pi}} \int d\tilde{y} \, (\tilde{y} - \tilde{x}_0)^2 \exp \left( -\frac{k}{2}(\tilde{y} - \tilde{x}_0)^2 \right)$$

The integral term, with its preceding square root, evaluates the variance of the Gaussian encoding distribution in the sensory space, which is equal to $1/k$, leaving

$$\mathbb{E}_{y|x_0}(y) - x_0 = \frac{1}{4k} \left( \frac{1}{f'(x_0)^2} \right)'_{x_0} = \frac{1}{4k} \left( \frac{1}{p(x)^q} \right)'_{x_0}$$

$$\implies b(x_0) = \mathbb{E}_{y|x_0}(\hat{x} - x_0) \approx \frac{1}{4k} \left( \frac{1}{p(x)^q} \right)'_{x_0} = \frac{1}{4} \left( \frac{1}{J(x_0)} \right)'_{x_0} \tag{32}$$

As indicated above, the original result follows immediately for $q = 2$ and $1/k = \sigma^2$.

$L_0$ **norm-minimizing MAP estimator** The mode of the $f$-normal posterior is equivalent to the mode of its log-posterior, thus

$$\frac{\partial \log p(x \mid y)}{\partial x} = \frac{p'(x)}{p(x)} - k(f(y) - f(x))f'(x)$$

Applying the same first-order Taylor expansion of $f(x)$ about $y$ (sic, ultimately we replace $f'(y)$ with $f'(x)$, so it would have been more reasonable to perform the converse expansion), we get

$$f(y) - f(x) \approx f'(y)(x - y)$$

$$\implies \frac{\partial \log p(x \mid y)}{\partial x} = \frac{p'(x)}{p(x)} - k(x - y)f'(y)f'(x)$$

$$= \frac{p'(x)}{p(x)} - k(x - y)p(y)^{q/2}p(x)^{q/2}$$

Setting the lefthand side equal to zero, we isolate the difference between the Bayesian estimator $\hat{x}$ and $y$ to be

$$\hat{x} - y = \frac{p'(\hat{x})}{p(\hat{x})kp(y)^{q/2}p(\hat{x})^{q/2}} \approx \frac{p'(x_0)}{kp(x_0)^{q+1}} = -\frac{1}{qk} \left( \frac{1}{p(x_0)^q} \right)'_{x_0} \tag{33}$$

$$\implies \mathbb{E}_{y|x_0}(\hat{x} - y) = -\frac{1}{qk} \left( \frac{1}{p(x_0)^q} \right)'_{x_0} = -\frac{1}{q} \left( \frac{1}{J(x_0)} \right)'_{x_0}$$

The bias under the MAP estimator becomes the composite sum of this equation with the posterior median above, such that

$$b(x_0) = \mathbb{E}_{y|x_0}(\hat{x} - x_0) = \mathbb{E}_{y|x_0}(\hat{x} - y) + \mathbb{E}_{y|x_0}(y - x_0)$$

$$= -\frac{1}{k} \left( \frac{1}{q} - \frac{1}{4} \right) \left( \frac{1}{p(x_0)^q} \right)'_{x_0} = -\left( \frac{1}{q} - \frac{1}{4} \right) \left( \frac{1}{J(x_0)} \right)'_{x_0} \tag{34}$$

Likewise, the original result with coefficient $-1/4$ follows immediately after resubstituting $q = 2$ and $1/k = \sigma^2$.

**Other $L_p$ Bayesian estimators** For any other $L_p$ norm-minimizing Bayesian risk minimization problem, we can recycle the Taylor expansion of $f^{-1}(\cdot)$ we developed above and repeat their derivations line-by-line to recover the same key relations wherein for some constant $A$,

$$b(x_0) = A\left(\frac{1}{J(x_0)}\right)'_{x_0} = A\left(\frac{1}{p(x_0)^q}\right)'_{x_0} \longrightarrow A\left(\frac{1}{p(x_0)^2}\right)'_{x_0} \quad \text{when } q = 2 \qquad (35)$$

Indeed, our concluding argument is that these derivations do not specially depend on infomax assumptions; they are a general property of an encoder constrained by some power function of the stimulus prior inferred from the environment, which we discuss in the following section.

## 3.2 Asymptotic equivalence of normal and $f$-normal encoding models

Although the assumptions and parameters seem different, it is straightforward to show that even this skewed $f$-normal noise model becomes a member of the same Gaussian class of encoding distributions in the high-SNR limit. Consider the first-order Taylor expansion of $f(y)$ about $f(x)$ employed when replicating of Wei and Stocker's results, which would state that $f(y) \approx f(x) + f'(x)(y - x)$ and subsequently that $f'(y) \approx f'(x)$. Together with the definitions of Fisher information for this $f$-normal parametrization $J(x) = kf'(x)^2$,

$$p(y \mid x) = f'(x)\sqrt{\frac{k}{2\pi}}\exp\left(-\frac{kf'(x)^2}{2}(y-x)^2\right) \longrightarrow \sqrt{\frac{kp(x)^q}{2\pi}}\exp\left(-\frac{kp(x)^q}{2}(y-x)^2\right)$$
$$(36)$$

where power-law efficient coding would dictate that $J(x) = kp(x)^q$. In this way, the Gaussian representation we present in our analysis of the Bayesian observer model in the high-SNR limit could supercede alternate parametrizations.