[Reviews · NeurIPS 2018]

Reviewer 1



This paper generalizes previous work by Wei & Stocker (2017), by showing that a general class of variational objectives (“power-law efficient codes”) can produce the same psychophysical law that relates perceptual bias and discriminability. The work derives a strong form of the law, that holds in the asymptotically high SNR range (in which the prior is dominated by the likelihood), and a “weak” form which is nonlinear but works in a regime of non-negligible noise. In both cases, the paper explicitly derives the proportionality constants for the psychophysical law. The framework of this manuscript is slightly different than that used originally by Wei & Stocker (2017), but the authors show (mostly in the Supplementary Material) that the results hold also for the original framework (in particular, the encoding differs between the two, even though the two have the same Fisher information). Comments: This paper is technically sound, very well written and clear despite the complexity of the subject. The paper takes off from previous work by Wei and Stocker (2017) which proposed a novel psychophysical law (a lawful relation between perceptual bias and discriminability), but provides substantial novel content in a derivation of an entire class of “power-law” codes that would induce the same relation. This is important as it implies that the proposed psychophysical law could emerge for many different kinds of codes, and not only for an information maximizing code as it was assumed in previous work. At the moment, it’s not clear where the ansatz for Eq. 21 comes from (I can guess it emerged from visual inspection of Figure 5C, and trying out a few functional forms). The authors might want to refer to Fig 5 here. (In fact, there is no reference to Fig 5 in the main text.) This is a strong submission with novel theoretical results that lead to experimental predictions for psychophysical experiments (e.g., in terms of the coefficient of proportionality between bias and discriminability), and novel ideas that elaborate on the concept of efficient coding. Typos: line 39: disriminability → discriminability line 138: differ → differs line 151.5 (unnumbered equation): remove either log on the left or exp on the right Eq 19: missing a minus sign in the argument of the exp line 315.5 (SI): in the first line of the equation, right-hand side, p(x) is missing line 430: minimzation → minimization line 432: wherein → therein line 444: supercede → supersede After authors' feedback: Nothing to add - well done.

Reviewer 2



The paper studies whether existing perceptual laws could hold under different, non-optimal, neural codings. In that respect, it generalizes previous results that showed that perceptual laws could arise from efficient neural coding. The paper is very well written, the formal parts very well developed and the material is finely organized. As a reader who hasn't had previous exposure with the literature cited, I get the impression that the authors expand substantially on previously results and it would be a good contribution to the conference. My main point of critique would be that the authors assume that the reader is a specialist, while a conference like NIPS is much more generalist. For instance, both bias and discriminability are loaded with meaning in the machine learning (bias-variance dilemma) and psychophysics/neuroscience literatures (different variants of signal detection theory). How do the terms employed by the authors connect with these commonly used terms? I get the impression that discriminability is used as in signal-detection theory in psychophysics and cognitive psychology. It would help a lot if the authors could give a precise example of the problem that the cognitive system they theorize about has to solve. In reference [1], on which the paper is based, the task solved and these terms were quite clearly explained. In the current publication, the meaning of these terms is, unfortunately, somewhat obscured.

Reviewer 3



The authors study efficient coding in neural populations and generalize previous results on a link between discriminability and perceptual bias to a larger family of population codes. In particular, they show that the relationship holds whenever Fisher information is allocated proportional to a power of the prior distribution and compute the constant of proportionality. This is interesting, as previous work had relied on optimal codes where Fisher information is related to the square of the prior. To this extent that I was able to check the derivations, they seem correct. This is an interesting contribution to the neural coding literature. Figure 2B: The three-dimensional figures does not help understanding. Figure 2D: I find the plot for linearization of the relationship not really convincing.